# Mushroom–Legume-Based Minced Meat: Physico-Chemical and Sensory Properties

**DOI:** 10.3390/foods12112094

**Published:** 2023-05-23

**Authors:** Md. Anisur Rahman Mazumder, Shanipa Sukchot, Piyawan Phonphimai, Sunantha Ketnawa, Manat Chaijan, Lutz Grossmann, Saroat Rawdkuen

**Affiliations:** 1Food Science and Technology Program, School of Agro-Industry, Mae Fah Luang University, Chiang Rai 57100, Thailand; anis_engg@bau.edu.bd (M.A.R.M.); sunantha.ketnawa@gmail.com (S.K.); 2Department of Food Technology and Rural Industries, Bangladesh Agricultural University, Mymensingh 2202, Bangladesh; 3Food Technology and Innovation Research Center of Excellence, School of Agricultural Technology and Food Industry, Walailak University, Nakhon Si Thammarat 80160, Thailand; mchaijan@gmail.com; 4Department of Food Science, University of Massachusetts Amherst, 102 Holdsworth Way, Amherst, MA 01002, USA; 5Unit of Innovative Food Packaging and Biomaterials, School of Agro-Industry, Mae Fah Luang University, Chiang Rai 57100, Thailand

**Keywords:** edible mushrooms, plant-based protein, chickpea, canola oil, beetroot extract, sensory attributes, alternative meat

## Abstract

A growing number of health-conscious consumers are looking for animal protein alternatives with similar texture, appearance, and flavor. However, research and development still needs to find alternative non-meat materials. The aim of this study was to develop a mushroom-based minced meat substitute (MMMS) from edible *Pleurotus sajor-caju* (PSC) mushrooms and optimize the concentration of chickpea flour (CF), beetroot extract, and canola oil. CF was used to improve the textural properties of the MMMS by mixing it with PSC mushrooms in ratios of 0:50, 12.5:37.5, 25:25, 37.5:12.5, and 50:0. Textural and sensory attributes suggest that PSC mushrooms to CF in a ratio of 37.5:12.5 had better textural properties, showing hardness of 2610 N and higher consumer acceptability with protein content up to 47%. Sensory analysis suggests that 5% (*w/w)* canola oil showed the most acceptable consumer acceptability compared to other concentrations. Color parameters indicate that 0.2% beetroot extract shows higher whiteness, less redness, and higher yellowness for both fresh and cooked MMMS. This research suggests that MMMS containing PSC, CF, canola oil, and beetroot extract could be a suitable alternative and sustainable food product which may lead to higher consumer adoption as a meat substitute.

## 1. Introduction

From the ancient period, meat has traditionally been considered as a necessary component of the human diet [1]. Consuming meat played a very crucial role in the development of prehistoric *Homo sapiens* and has become a dominant food item for the human diet in many regions of the world, with unforeseen consequences [2]. Chicken, beef, mutton, and pork are the most popular items throughout the world, and the countries with the highest yearly meat consumption per capita are the United States and Australia [3]. Meat demand has risen by 58% globally during the last 20 years, owing to population growth and strong economic progress [4]. There have been a few concerns about the harmful consequences of meat intake on human health and the inefficiency of raw and processed meat production when compared to agricultural crop production [5]. Meat production and consumption have been associated with human health issues, including an increased risk of zoonoses, chronic illnesses, and health issues connected to air pollution [6,7]. These detrimental consequences of meat in sustainable development are causing a rising number of people to turn vegetarian or flexitarian, which means that they aim to minimize their meat consumption as much as possible [8]. In addition, because of the pandemics of COVID-19 and the African swine fever virus, scientists and researchers have started to reevaluate the safety of low-temperature meat supply chains [9]. An analysis of 90 dietary recommendations from around the world found that 37% suggested substituting meat protein for vegetable protein [10] which shows the importance of establishing a sustainable source of human protein nutrition. To address these issues, food scientists and the food industry are exploring ways to offer meat alternatives of plant and mushroom origin with the aim of mimicking original animal tissue in terms of texture, aroma, taste, and appearance [11,12]. According to predictions by Union de Banques Suisses (UBS) [13], the market for non-animal meat either from plants and/or edible fungi or insects will increase from USD 4.6 billion in the year 2018 to USD 85 billion by the year 2030, and as a notable milestone, reach USD 30.9 billion by the year 2026 [14]. This huge new market seems to be well suited for the development and invention of new meat alternatives.

The food research community is now studying three main meat substitutes: plant-based meat (developed from plant proteins mainly using mechanical structural techniques) [15] mushroom-based meat, using mainly ascomycetes or basidiomycetes, and cultured meat based on animal tissue engineering [16]. There is an increase in interest in meat analogs developed from more environmentally friendly non-animal proteins [17]. However, a lot of plant-based (PB) meat substitutes are available in local supermarkets, whereas only a few mushroom-based meat alternatives have been released (for example, Quorn products).

Most frequently, soy and wheat gluten are used as potential sources of protein for the processing of meat analogs [18]. Nevertheless, sources of protein from mushrooms as well as legumes such as peas, faba beans, kidney beans, and others have been utilized for the development of meat substitutes [19]. Mushrooms are attractive and highly valued due to their unique flavors and textures, their high nutritional value, with 4 g/100 g protein (depending on the species), and high dietary fiber content, which is mainly composed of β-glucan [20]. Edible mushrooms are used in the processing of meat analogs for human consumption because they are a rich source of macronutrients (such as protein and fiber) and micronutrients (such as vitamins and minerals). Edible mushrooms contain low amounts of fat and sodium, and possess low energy content [21,22,23,24]. Mushrooms also contain a number of bioactive chemicals, including proteoglycans, phenolic compounds, terpenes, and lectins [20].

Mushrooms have yet to be introduced as a raw material for the processing of minced meat substitutes. Some earlier studies by Yuan et al. [25] processed a fibrous meat analog utilizing thermo-extrusion and developed different formulas for manufacturing sausage analogs. However, extrusion requires high-temperature and shear conditions and might not be readily available everywhere. Thus, there is a need to develop meat alternatives based on mushrooms that do not require the use of extrusion techniques. Moreover, mushrooms contribute to the formation of primary sensory attributes including taste, texture, and appearance, which might be negatively influenced by extrusion. Recently, a PB emulsion-type sausage (ES) was developed from gray oyster mushrooms and chickpeas by Mazumder et al. [26]. In comparison to commercial sausages, the ES had more protein (36% on a dry basis), lower cooking loss (4.08%) and purge loss (3.45%), stronger emulsion stability, and improved higher acceptance [26]. In addition, gray oyster mushrooms and chickpea flour (CF) may be suitable substitutes for soy protein in PB meat products [26]. Consequently, the goal of this project is to develop mushroom-based minced meat substitutes (MMMSs) that may be claimed as clean-label products and to develop a value-added meat substitute that might satisfy customer demand. The objectives of this study are to (*i*) develop an MMMS using mushrooms, CF, beetroot extract, and canola oil, (*ii*) optimize the formulation using PSC mushrooms, CF, beetroot extract, and canola oil, and (*iii*) reveal the textural as well as the sensory properties of the optimized and developed formulations.

## 2. Materials and Methods

### 2.1. Materials

The edible and raw *Pleurotus sajor-caju* (PSC) mushrooms were bought from the local fresh market (Bandu, Chiang Rai, Thailand). Canola oil extract was purchased from the local supermarket (BigC, Chiang Rai, Thailand). Chickpea flour (moisture 11.85 ± 0.01, protein 22.18 ± 0.09, fat 5.52 ± 0.07, ash 2.61 ± 0.01, and carbohydrate content 69.70 ± 0.05, wt%, d.b.) was purchased from Huglamool Farm (Amnat Charoen, Thailand). Beetroot extract was purchased from the Narah herb company (Chiang Mai, Thailand). Vital wheat gluten (moisture 8.84 ± 0.01, and protein content 87.94 ± 0.39, wt%, d.b.), and soy protein isolate (moisture 8.93 ± 0.02, and protein content 90.17 ± 0.17, wt%, d.b.) were purchased from Union Science Co., Ltd. (Chiang Mai, Thailand). Yeast extract was purchased from Thai Food and Chemical Co., Ltd. (Bangkok, Thailand).

### 2.2. Mushroom Preparation

At first, the PSC raw mushrooms were washed with potable water several times to remove foreign materials. The cleaned mushrooms were blanched at 100 °C for 5 min to ensure their storability before mincing them using a meat mincer (SIR1-TC12E, SEVENFIVE DISTRIBUTOR Co., Ltd., Nonthaburi, Thailand) followed by mechanical drying at 60 °C in a cabinet tray dryer (BP-80, Kluay Nam Tai, Bangkok, Thailand). The final moisture content in the mushrooms was 65%. The dried mushrooms were vacuum-packed and kept at −20 °C until further use.

### 2.3. MMMS Preparation

#### 2.3.1. Base Formulation

Frozen mushrooms were thawed in the refrigerator overnight at 4 °C. Dried mushrooms (25%, *w/w)* were mixed with chickpea flour (25%, *w*/*w*), vital wheat gluten (4.8%, *w*/*w*), distilled water (28%, *w*/*w*), soy protein isolate (10%, *w*/*w*), canola oil (5%, *w*/*w*), beetroot extract (0.2%, *w*/*w*), and yeast extract (2%, *w*/*w*) (Table 1). Figure 1 illustrates the processing flow diagram of the MMMS. These ingredients were selected to accurately mimic the taste and appearance of minced meat. All ingredients were blended in a mixing bowl until they were homogeneously mixed. The mixture was placed into a meat mincer (SIR1-TC12E, SEVENFIVE DISTRIBUTOR Co., Ltd., Nonthaburi, Thailand) to form the typical minced meat shape, and transferred into a baking oven at 150 °C for half an hour [27]. All of the samples were individually vacuum-packed for further analysis. 

#### 2.3.2. Formulation Optimization

In order to further optimize the MMMS, three treatments were designed: (i) the effect of different concentrations (0, 12.5, 25, 37.5, and 50% *w*/*w*) of chickpea flour on the textural and sensory attributes of the MMMS (Table 2), (ii) the effect of different concentrations of beetroot extract (0, 0.2, 0.4, 0.6, 0.8, and 1.0% *w*/*w*) on the color parameters and sensory quality of the MMMS (Table 3), and (iii) the effect of canola oil concentrations (0, 1, 2, 3, 4, and 5% *w/w)* on the sensory attributes of the mushroom-based MMMS (Table 4).

### 2.4. Chemical Analysis of Mushroom-Based MMMS

#### 2.4.1. Proximate Composition Analysis

Using the 2019 AOAC recommendations, proximate composition, including moisture, ash, protein, and fat content, was determined. Moisture content was assessed by placing 3 to 5 g of the sample into a convection oven at 105 °C for at least 16 h [28]. Ash content was determined via the combustion of a sample in a muffle furnace for 6 h at 525 °C [29]. Protein content was measured using the Kjeldahl method, utilizing the nitrogen-to-protein conversion ratio of 5.99 [30]. The Soxhlet extraction technique was used to determine the fat content [31]. The total carbohydrate was calculated in accordance with the FAO guidelines, as is the remainder [32]. All values were measured three times and the results were presented as means ± standard deviation.

#### 2.4.2. Determination of Total Dietary Fiber (TDF)

The amount of TDF was measured using an enzymatic–gravimetric technique [33]. Duplicate portions of defatted and dried samples were gelatinized and partially digested by α-amylase before being enzymatically digested with protease and amyloglucosidase to remove the protein and starch from the sample. Acetate buffer (5 mL, 0.1 M, pH 5.0) containing 100 µL thermostable α-amylase was mixed with about 300 g of the MMMS before being incubated at 96°C for 1 h in a tightly sealed container. The suspension was then incubated at 60 °C for 4 h after 400 µL of amyloglucosidase and protease was added. Subsequently, 80% aqueous ethanol was added to precipitate soluble fibers. The suspension was centrifuged at 2000 rpm for 20 min to collect the total fiber. The residue was washed with ethanol and acetone, followed by drying and weighing. A portion of the residue was tested for ash, and another portion was tested for protein. TDF was computed as a percentage of the initial sample weight by subtracting the weight of the residue from the weight of the protein and ash. All values were measured three times and the results were presented as means ± standard deviation.

#### 2.4.3. Determination of Amino Acids of PSC

The sample for amino acid analysis was prepared in accordance with the procedure described by Borokini et al. [34]. Fresh PSC mushrooms (20 g) were precisely weighed and pulverized in a blender with 100 mL of phosphate buffer containing 2% sodium dodecyl sulfate (SDS). The suspension was filtered using a double-layered cheesecloth. The filtrate was precipitated using an ammonium sulphate salt precipitation technique at 65% saturation. For amino acid analysis, the proteins were pelleted via centrifugation, dialyzed to concentrate them, and then freeze-dried. A rotary evaporator was used to hydrolyze and evaporate 4 g of protein isolate. The amino acid composition of the fresh PSC mushrooms was analyzed in the Central Laboratory, Chiang Mai, Thailand, using an in-house method TE-CH-372 based on the Official Journal of the European Journal of Communities, L 257/16, and the result was reported as g/100 g sample.

#### 2.4.4. SDS Polyacrylamide Gel Electrophoresis (SDS-PAGE)

SDS-PAGE was used to observe the protein patterns of the MMMS. The samples were boiled for 3 min after being mixed with sample buffer (0.5 M Tris-HCl, pH 6.8 containing 4% SDS, 20% glycerol, 0.03% bromophenol blue with/without 10% DTT) in a 1:1 ratio. The protein dye was loaded into Roti-PAGE Gradient (4–20%) precast gels and run in an electrophoresis tank filled with buffer solution at a constant current of 60 mA using a Biostep^®^ GmbH power supply (Burkhardtsdorf, Germany). The gel was stained overnight in a staining solution (Coomassie Blue R-250 methanol-acetic acid) with moderate shaking at 50 rpm. The gel was de-stained using de-staining solutions I and II (methanol-acetic acid–water) until the background was clear, followed by drying.

### 2.5. Physical Analysis of MMMS

#### 2.5.1. Textural Profile Analysis (TPA)

The TPA of the MMMS was measured using the methods described by Tasnim et al. [35], with modifications. For the sample preparation, the MMMS was formed into a patty using a Petri dish to transform it into a round-shaped structure (1.5 cm × 5.0 cm) (height x length). TPA was determined using a TA.XT plus Texture Analyzer (Surrey, UK). TPA was performed using a cylindrical probe (SMSP/36R, cylinder diameter = 36 mm). The pre-test speed was 1 mm/s, the test speed was 5 mm/s, the post-test speed was 5 mm/s, the strain was 50%, the trigger force was 10 g, and the time interval between the two compressions was 5 s. The TPA was computed using EXPONENT CONNECT^®^ software (Stable Microsystems, Surrey, UK) as hardness (N), chewiness (N), springiness (mm), cohesiveness, and gumminess. All values were measured five times and the results were presented as means ± standard deviation.

#### 2.5.2. Cooking Loss

The cooking loss was determined using five different MMMS samples and by calculating the ratio of weight before and after cooking. The MMMS was soaked in distilled water in a ratio of 1:1 (*w/v*) for 1 h (soaked/uncooked) followed by cooking for 15 min in a pan without oil. It was then allowed to cool at room temperature. The following Equation (1) was used to calculate the cooking loss [36]:(1)$$\mathrm{Cooking}\;\mathrm{loss}\;(\%)=\frac{\mathrm{Raw}\;\mathrm{MMMS}\;\mathrm{weight}\;(g)\;-\;\mathrm{Cooked}\;\mathrm{MMMS}\;\mathrm{weight}\;(g)}{\mathrm{Raw}\;\mathrm{MMMS}\;\mathrm{weight}\;(g)}\;\times \;100\;$$

#### 2.5.3. Color Determination

A colorimeter (Hunter Lab/colorQuest XE, Reston, Color Global, Bangkok, Thailand), utilizing a 10° standard observer and illuminant D65, was used to measure the color of the MMMS. A standard white plate was used to calibrate the colorimeter. Ten randomly selected samples were used to record the CIE L*, a*, and b* values of the samples. The lightness was indicated by the L*, which ranged from black (L* = 0) to white (L* = 100). The a* stands for the red (positive a*) and green spectra (negative a*). The positive b* represents yellowness and the negative b* indicates blueness. These characteristics were also utilized to calculate Δ*E* and whiteness [37]. A Δ*E* >2.0 is considered to be a color difference.

### 2.6. Sensory Analysis

A 9-point hedonic scale (1 = extremely dislike and 9 = extremely like) was used to evaluate the sensory properties of the MMMS [38]. Sensory analysis was carried out in the Food Sensory Lab (S4) (Mae Fah Luang University, Chiang Rai, Thailand) with ethical approval (protocol no.: EC22177-14) for consumer testing. Sensory analysis was permitted by Mae Fah Luang University, Chiang Rai, Thailand (CoE158/2022). Samples were evaluated by untrained panelists in the following numbers: 46 (23 female and 23 male) for base formulation, 46 (23 female and 23 male) for the experiment of different concentrations of chickpea flour and PSC, 34 (17 female and 17 male) for the experiment of different concentrations of canola oil, 35 (18 female and 17 male) for the experiment of different concentrations of beetroot extract, and 120 (60 female and 60 male) for the final formulation. The age range of the untrained panelists was 18–75 from Chiang Rai province, Thailand. Panelists were chosen from both regular consumers of PB meat and non-vegans. Each study of the MMMS was conducted for sensory attributes including appearance, texture, odor, taste, and overall acceptability. To prevent the influence of sample order presentation, samples were provided to panelists once at a time. Between sampling, panelists were instructed to drink water to cleanse their palate. The MMMS sensory session was conducted at 25 °C in separate rooms (individual cabins) under controlled environmental conditions with white light (300 lx) and 54% relative humidity. Furthermore, to minimize the impact of shock, all panelists were informed in advance that novel items were being developed to replace conventional animal minced meat.

### 2.7. Statistical Analysis

Each set of data was collected in triplicate except for color parameter and TPA, and was reported as mean ± standard deviation. The Statistical program for Agricultural Research (STAR) software program (International Rice Research Institute, Manila, the Philippines) was used to analyze all of the data using one-way analysis of variance (ANOVA). The 95% confidence level (*p* < 0.05) was considered to be statistically significant among different samples. 

## 3. Results and Discussion

### 3.1. Properties of Pleurotus sajor-caju Mushrooms

Before the MMMS was prepared, the PSCs were analyzed for their morphological attributes and composition. Table 5 shows that PSCs have the highest essential amino acids (except for lysine) when compared with the requirement pattern in protein (%) for adults). Moreover, overall considerable sensory characteristics were observed for PSCs in MMMS formulations. For this reason, PSCs were chosen to prepare and optimize the MMMS formulations. Before preparation, the PSCs were analyzed for ash, protein, fat, dietary fiber, and amino acid content (Table 5).

Dietary fiber is the most abundant component of PSC mushrooms, followed by proteins and other carbohydrates. This dietary fiber is mainly composed of β-glucan, which was present in the PSCs at 25.72 g/100 g dry weight (DW). β-glucan stimulates the host immune system to protect the host against bacterial, viral, fungal, or parasitic infections [39]. By attaching to the receptor (dectin-1) of the host cells, β-glucan stimulates macrophages, neutrophils, and natural killer cells [40,41]. On a final note, PSCs contain considerable amounts of indispensable amino acids with many of them found at higher concentrations than those recommended by the FAO for different age groups of adults. However, actual bioavailability data and PDCAAS/DIAAS values are currently missing for this mushroom in order to draw a final conclusion on the protein quality.

**Table 5 foods-12-02094-t005:** Morphological characteristics and nutritional properties of *Pleurotus sajor-caju* mushrooms.

Properties	*Pleurotus* *sajor-caju*	%RP
Morphology		
Size	Stalk length: 2.8 cm; stalk diameter: 1.1 cm; diameter of cap = 6 cm.	-
Shape	Cap is a fleshy shell or is spatula-shaped (pileus); stipe (stalk) is lateral (short or long) or central; gills (lamellae) are lengthy ridges and furrow underneath the pileus.	-
Weight/Age	28 to 35 g/25 to 30 days	-
Nutritional properties (% dry weight)		
Ash	7.85 ± 0.09	-
Protein	24.79 ± 0.9	-
Fat	1.15 ± 0.08	-
Dietary fiber	43.75 ± 3.50	-
Essential amino acids (g/100 g sample)		
Histidine	2.20	1.9
Lysine	4.94	5.08
Isoleucine	4.61	2.8
Leucine	7.17	6.6
Tryptophan	1.13	-
Phenylalanine	6.05	6.3 ^a^
Threonine	4.74	3.4
Methionine	1.59	2.5 ^b^
Valine	5.07	3.5

%RP = requirement pattern in protein (%) for adults [42], ^a^ = Phenylalanine with tyrosine, ^b^ = Cysteine with methionine.

### 3.2. Properties of Base Formulation

After the main components of the PSC mushrooms were identified, an MMMS was prepared using PSCs as the main ingredient (50%, *w*/*w*). The result suggests that the moisture content of the PSC-based MMMS was 28.39 ± 0.17% (Table 6). The protein content of the PSC formulation was 41.99 ± 0.55%, which was considerably higher than the initial protein content of the mushrooms, as well as regular pork minced meat (Table 7). This can be attributed to the formulation that contained CF, wheat gluten, and soy protein. The sensory attributes provide information about the overall acceptability, appearance, aroma, color, taste, and texture of the MMMS formulated with the PSCs. The results showed that the overall acceptability of the MMMS base formulation is in the range of “Like Slightly”. This is not surprising since this is a new type of food and many consumers reject foods when they try it for the first time [43,44]. Nonetheless, the acceptability was already high using the base formulation, but especially taste and smell were ranked rather low. This might be due to the aroma compounds that are typically found in mushrooms, such as 1-octen-3-ol, hexadecanoic acid, and octadecenoic acid [45,46,47]. These compounds are not commonly found in real meat products and therefore might have caused an adverse rating in such a product that is designed to replace real animal food. However, as the base MMMS formulation was overall positively evaluated by the panelists, further formulation improvements were investigated which will be discussed in the following sections.

### 3.3. Effect of Concentrations of Chickpea Flour (CF)

CF is a commonly used food ingredient and is also regularly utilized as a binding and texturizing agent in meat alternatives [48,49]. Typically, it is observed that the hardness of food matrices is increased when CF is added to the formulations [50]. As a result, the goal of these experiments was to elucidate the effect of increasing CF concentration on the textural and sensory properties of MMMS. For this, the PSCs were replaced with CF at concentrations from 0 to 50% (Table 2).

The moisture and protein content (dry basis, g/100 g) of the PSC-mushroom-based MMMS are shown in Table 7. In particular, moisture content was significantly (*p* < 0.05) increased while increasing the PSC mushrooms in the MMMS. This might be due to the residual moisture content of PSC mushrooms (MC = 65%) after drying. The moisture content of the PSC-mushroom-based MMS was the highest in the PSCs to CF ratio (50:0) and significantly different from the other samples (*p* < 0.05). However, the moisture content of the PSC-mushroom-based MMS was much lower than the minced beef and pork, which were 61 and 53%, respectively [51,52]. The moisture analysis of this study indicates that the moisture content of the MMMS was much lower (*p* < 0.05) than that of pork minced meat (PMM) (60.10%) (Table 7). The highest protein content was observed for the MMMS with PSCs to CF ratios of 37.5:12.5 and 50:0 and the values were 47.03 and 47.59%, respectively. The MMMS with PSCs to CF ratios of 37.5:12.5 and 50:0 did not show any significant difference (*p* > 0.05) in terms of protein content. Table 7 suggests that the protein content of the MMMS increased with the increase in mushroom content. These results show that PSC-based MMMSs can contribute considerably to the protein supply in the human diet and future protein quality assessments should be carried out to analyze the bioavailability of the amino acids [53]. The results suggest that the protein content of the PSC-mushroom-based MMMS was higher than that of minced beef (25.53%) and minced pork (25.7%) [51,52]. This study also revealed that the protein content of the MMMS was higher than that of PMM (20.17) (Table 7). However, the cooking loss in protein for both the MMMS and PMM showed similar trends.

In the next step, the change in the texture of the MMMS with increasing CF was analyzed. TPA measurements showed that the addition of CF to the MMMS had a considerable influence on its textural attributes. Hardness values ranging from 1983.35 N (PSC:CF = 50:0) to 9441.01 N (PSC:CF = 0:50), springiness values from 0.65 mm (PSC:CF = 0:50) to 0.90 mm (PSC:CF = 50:0), and cohesiveness values from 0.35 (PSC:CF = 0:50) to 0.63 (PSC:CF = 50:0) were detected. Hardness and chewiness showed similar patterns among treatments, with 50% CF showing the highest value for both hardness and chewiness. The treatments with the highest PSC concentration, 37.5% and 50% from mushrooms, exhibited low hardness values (2610.23 and 1983.355 N, respectively) (Table 7). The results indicated that this treatment significantly reduced the force required to compress the sample, which can have important consequences for the mouth feel of a product. The reason for this is most likely the higher porosity induced by the higher concentration of PSCs and the lower crosslinking with water-soluble proteins, which is expected to increase the hardness in samples containing higher amounts of CF. The MMMS with 50% CF showed the highest hardness due to an increase in bulk density, decreased porosity, and decreased water-holding capacity [54].

The findings revealed that the MMMS made with an increasing CF concentration decreased the springiness of the sample and that the MMMS with the pure PSC mushrooms had the highest potential to regain its original dimension following compression. This shows a high degree of protein texturization that allows for energy conservation and thus elastic deformation, presumably in the form of disulfide bond cross-links [53]. The 0% CF (50% PSC mushroom) MMMS had sponge-like springiness following hydration, which, however, was not a meat-like texture. All treatments with additional CF had significantly less springiness, which indicated a strong influence of the presence of starch in the formulation that contributed to changing the textural properties of the MMMS matrix [54]. A low springiness value, on the other hand, suggests that the material was plastically deformed [55]. Moreover, the maximum chewiness was observed in formulations with 50% CF (0% PSC mushrooms). The result corresponds with the hardness value. While chewiness is a computed measure that is partially derived from hardness and springiness, hardness predominates due to its higher values when compared to the other treatments. Table 7 suggests that chewiness was dramatically reduced by more than 50% with the addition of 12.5 to 37.5% PSC mushrooms. Texture analysis suggests that the PMM had better (*p* < 0.05) textural properties than the MMMS. The hardness and chewiness values were higher in the MMMS than the PMM. The hardness value was 1983.35–9441.01 N for the MMMS, whereas the value was 1925.35 N for the PMM. Similar trends were observed for cooked MMMSs and PMMs (Table 10). However, textural parameters were significantly (*p* < 0.05) better in the MMMS than the commercial plant-based minced meat (CPBMM) with lower hardness and chewiness for both fresh and cooked samples (Table 10). For meat analog products to be as widely accepted by consumers as possible, textural characteristics should, however, closely resemble those found in meat products. The TPA measurements revealed that controlling the protein-to-starch ratio by optimizing the CF and PSC content can be a crucial factor in determining this desirable quality attributes. Due to the negative effects of decreased chewiness, a higher value of springiness without sufficient hardness, as in the case of 50% PSC mushroom treatment, may reduce consumer acceptability. In light of the aforementioned discussion, it can be anticipated that an MMMS product made with 12.5% CF and 37.5% PSC mushrooms will have the highest level of customer acceptance (Table 7). To elucidate the answer to this question, a sensory test was carried out.

#### Sensory Properties of *Pleurotus sajor-caju* Mushroom-Based Minced Meat Substitutes

The composition analysis suggested that the MMMS with PSCs to CF ratios of 37.5:12.5 and 50:0 had the highest protein concentration, but the MMMS with 50% PSC-mushrooms was most likely less suited for food applications because of the adverse textural attributes revealed in the TPA measurements. As a result, consumer preference testing was conducted via sensory evaluation of 46 untrained panelists. As already projected in the TPA measurements, the sensory analysis suggested that 37.5% PSC mushrooms and 12.5% CF exhibited the highest overall acceptability, followed by other MMMSs. The appearance ratings of the MMMSs also suggest that the 37.5% PSC mushroom and 12.5% CF-based MMMS is rated the best among all ratios. Similarly, the 37.5% PSC mushroom and 12.5% CF MMMS showed better texture acceptability according to the panelists following other samples. The MMMS containing 50% CF showed the least acceptability by the consumer due to a hard texture and high chewiness. The consumer acceptability of meat substitutes depends on the taste, color, and flavor of the products [43]. The overall acceptability of the MMMS made from PSC mushrooms and CF in ratios of 0:50, 12.5:37.5, 25:25, 37.5:12.5, and 50:0 was between dislike slightly (consumer scores above 4.0) and like moderately (scores above 7.0). The MMMS with 37.5% PSC mushrooms and 12.5% CF was the best according to the sensory analysis, and showed that it was moderately liked (scores 7.24) by the consumers, whereas the PMM was liked very much (scores 8.04). However, Table 7 indicated that the 37.5% PSC mushroom and 12.5% CF-based MMMS showed the highest textural and sensory acceptability. As a result, an MMMS formulation containing 37.5% PSC mushrooms and 12.5% CF was selected for the following experiments. Consumers who eat meat have the tendency to compare meat substitutes with traditional beef, mutton, or pork. Customers have been advised to eat less for better health and environmental reasons. A possible solution is to replace animal meat with plant-protein- and mushroom-based substitutes; however, consumer acceptance of these products is still low, possibly due to taste and flavor [44]. As a result, it is crucial to determine the sensory characteristics that must be optimized to increase palatability [43]. In our study, more than two thirds of consumers were classified as omnivores, implying that meat played a significant role in their daily diet. However, the purchase and/or likeability of plant/mushroom-based meat substitutes vary from country to country. For example, (i) those who are particularly connected to meat in the United States are less likely to purchase or like PB meat substitutes. Appeal, excitement, and low disgust were all attitudinal predictors of purchase intent. (ii) In China, women are more prone than males to buying PB meat substitutes. Meat eaters are substantially more likely to purchase PB meat alternatives than vegetarians and vegans. A higher meat attachment indicates a higher chance of purchasing. Higher familiarity and less food neophobia are predictive of purchase intent. (iii) Omnivores and individuals who consume more meat tend to consume PB meat substitutes more frequently in India. Consumers from higher socioeconomic status categories, as well as those who are highly educated and more liberal ideologically, exhibited a greater interest in PB meat alternatives. Food neophobia indicated a lower purchase intent, but familiarity with the products predicted higher buying intent. In India, perceived sustainability, enthusiasm, necessity, and goodness all predicted PB meat substitute purchase intent [56]. One third (or fewer) of respondents are identified as vegetarians, vegans, or pescatarians. An increased focus on environmental and health-related factors might aid in the adoption of PB meat substitutes. Despite a few obstacles, there is undeniably a large market potential for PB meat substitutes, especially MMMSs, which is projected to grow in the future as environmental and health awareness grow [43]. 

### 3.4. Effect of Canola Oil on Sensory Characteristics

The base formulation contained 5% canola oil, which may adversely affect the purchasing decisions of consumers who are looking for a low-fat product. For this reason, the effect of decreasing the oil content was investigated. For this part of the study, 37.5% *w*/*w* PSC mushrooms and 12.5% *w*/*w* CF were utilized as these were determined to be the optimum concentrations in the previous section. From there, the canola oil concentration was changed to the range of 0–5%, and sensory analysis was investigated. The result suggested that the formulation containing 5% *w*/*w* canola oil significantly (*p* < 0.05) exhibited the highest consumer acceptability, whereas that with 0% canola oil exhibited the lowest consumer acceptability. Although those with 1, 2, and 3% *w*/*w* canola oil had similarities based on appearance, texture, juiciness, and overall acceptability, the sensory acceptance of these formulations was lower than for matrices containing 5% of canola oil (Table 8). In general, that with 5% *w*/*w* canola oil exhibited the highest score and that with 0% canola oil exhibited the lowest score for appearance, texture, and juiciness, from 34 untrained panelists. This is consistent with the findings of previous research, such as those published by Wi et al. [57], who utilized 15–35% canola oil for the processing of meat analogs and found that the addition of canola oil reduces cooking loss, increases water holding capacity, and improves sensory characteristics. In addition, Selani et al. [58] discovered that using canola oil as a fat substitute in a beef burger improved the cohesion and springiness in its sensory attributes. To reduce the quantities of saturated fatty acids and cholesterol in some meat substitutes, animal fats are typically substituted with vegetable oils, including olive oil [59,60], palm oil [61], canola oil, and coconut oil [62,63]. Various amounts of oil are used, depending on the raw materials, to give meat alternatives a more meat-like texture and to enhance their flavor, juicy quality, tenderness, and sensory qualities [26]. Canola oil is often regarded as a healthy oil due to its low saturated fat content (7%), which further supports the utilization of canola oil in MMMS formulations, since canola oil includes omega-6 and omega-3 fatty acids in a ratio of 2:1, which is thought to lower low-density lipoprotein (LDL) and total cholesterol levels [64].

### 3.5. Effect of Beetroot Extract on Color and Sensory Characteristics

The visual appearance of food products considerably influences consumer acceptability [65]. After establishing the optimum texture, attention should be given to the color or changes in color during product processing and cooking. Beetroot extract is often used as a natural coloring agent to mimic the red-pink appearance of uncooked meat [66]. For this reason, beetroot extract was chosen as a coloring agent to enhance the appearance of the MMMS, which appears brownish without a colorant. Moreover, beetroot extracts undergo color changes as a result of thermal deterioration and thereby mimic the natural color change that occurs when cooking meat [67]. For these experiments, 37.5% *w*/*w* PSC mushrooms, 12.5% *w*/*w* CF, and 5% *w*/*w* canola oil were used, and the beetroot extract concentration varied from 0% *w*/*w* to 1.0% *w*/*w*. Table 9 shows the results of the color measurements before and after cooking in a pan with low-flame heat for 8 to 10 min until browned, as well as for the sensory trials for these formulations. Low concentrations of beetroot extract in both fresh and cooked samples had a significantly high (*p* < 0.05) lightness (L*) value, positive a* (redness), and positive b* value (yellowness). Moreover, the a* value increased with the addition of beetroot extract, which was expected due to the high coloring power of the betanin found in this ingredient [67]. The a* value then decreased upon cooking due to the thermal degradation of betanin [67]. This was in line with the increase in the L* values of the cooked MMMS at each concentration when compared to the fresh samples (Table 9). It is frequently challenging to mimic the color change that takes place while cooking meat. Therefore, it is necessary to replace or mix a heat-stable natural or artificial food grade color or combination of colors that enable a color change similar to animal meat during cooking, grilling, or frying. For example, myoglobin denaturation results in a color change in beef muscle tissue at around 75 °C [67,68]. To mimic this color pattern in meat analogs, beetroot extract and betanin are suggested to be added as food additives to mimic a raw meat color [69,70,71], and exhibit color changes as a result of thermal degradation [72]. Beetroot extract is also added as a food colorant for the burger formulation of Beyond Meat^TM^ [73,74]. Color analysis indicates that a fresh MMMS without beetroot extract shows higher L* (lightness), lower a* (redness), and higher b* (yellowness) values than those with other concentrations of beetroot extract. A similar trend was observed for cooked MMMSs as well (Table 9). However, research has revealed that beetroot extract is often used as a natural coloring agent to mimic the red-pink appearance of uncooked meat [66]. As a result, sensory analysis of both fresh and cooked MMMSs was determined to find the best concentration of beetroot extract.

The effect of beetroot extract and cooking on the physical appearance of the MMMS is illustrated in Figure 2 and sensory analysis scores are in Table 9. In general, the inclusion of beetroot extract enhanced the overall acceptance scores of the MMMS. The overall acceptability, appearance, and fresh and cooked color acceptance of each sample were significantly different (*p* < 0.05) from each other. The analysis suggested that increasing the beetroot extract content in the MMMS decreases the consumer preference possibly because the overall redness is too intense and may be perceived as being artificial. However, consumer preferences were very low (*p* < 0.5) for the MMMS without beetroot extract. The sensory analysis suggests that an optimum quantity of beetroot extract is necessary for the processing of the MMMS. In fact, a higher concentration of beetroot extract resulted in a dark pink color (Figure 2 and two-way ANOVA suggested that 0.2% *w*/*w* beetroot extract exhibited significantly higher appearance acceptability. There were no significant differences (*p* > 0.5) in the texture of the MMMS after the addition of different concentrations of beetroot extract reported by the panel. The cooked MMMS showed significantly (*p* < 0.05) higher aroma scores than the aroma scores of the fresh MMMS. The result indicates that the cooking method is not responsible for the development of off-flavor and may even contribute to flavor development. This is consistent with Sucu and Turp’s [75] findings that the cooking of fermented sausages with beetroot powder (0.12, 0.24, and 0.35%) significantly (*p* < 0.05) increased the inner color scores. Moreover, other studies with fresh pork sausage that included additional beetroot extract (1 mL/kg) had a higher consumer acceptance score than the control (no colorant) sausages [76]. Overall, these findings indicate that adding beetroot extract as a natural colorant to an MMMS improves its sensory qualities and that 0.2% *w*/*w* beetroot extract is enough to achieve high sensory acceptance.

### 3.6. Analysis of the Optimized Pleurotus sajor-caju Mushroom-Based Minced Meat Substitute

The previous results revealed that the PSC MMMS based on 37.5% PSC mushrooms, 12.5% *w*/*w* chickpea flour, 0.2% *w*/*w* beetroot extract, and 5% *w*/*w* canola oil shows high consumer acceptance based on color, texture, and sensory attributes. The goal of this last section was to thoroughly evaluate the properties of this optimized formulation for both fresh and cooked MMMSs. In this section, the optimized MMMS was compared with CPBMM and PMM.

#### 3.6.1. Appearance and Textural Properties

The appearance of the optimized MMMS, CPBMM, and PMM recipes in terms of fresh and cooked is shown in Figure 3. It looks similar to CPBMM in terms of coarse particle size. The optimized MMMS shows a granular size of about 3–4 mm length and 2–2.5 mm width, whereas CPBMM shows longer granules of about 5–6 mm length and 3–3.5 mm width. The color of the MMMS seems to be reddish and mixed brown. While cooking, the color disappears and is discolored to brown. The color of optimized MMMS is comparable to the color of CPBMM (Figure 3) for both fresh and cooked, which correlates to the color parameters shown in Table 10. However, the color of cooked PMM was more whitish than both cooked MMMS and CPBMM (Figure 3 and Table 10).

**Table 10 foods-12-02094-t010:** Nutritional, physico-chemical, textural, and sensory properties of the *Pleurotus sajor-caju* mushroom-based minced meat substitute.

Properties	Fresh MMMS	Cooked MMMS	Fresh CPBMM	Cooked CPBMM	Fresh PMM	Cooked PMM
Nutritional composition						
Moisture (%)	12.06 ± 0.26 ^c^	9.78 ± 0.66 ^be^	10.29 ± 0.35 ^d^	7.89 ± 0.55 ^f^	60.10 ± 0.25 ^a^	55.27 ± 0.55 ^b^
Protein (%)	47.90 ± 0.74 ^a^	45.06 ± 0.90 ^b^	47.75 ± 0.50 ^a^	44.73 ± 0.80 ^b^	20.17 ± 0.70 ^c^	18.05 ± 0.25 ^d^
Fat (%)	12.51 ± 0.66 ^c^	10.76 ± 0.40 ^d^	4.19 ± 0.20 ^e^	3.76 ± 0.22 ^f^	17.80 ± 1.50 ^a^	16.79 ± 0.33 ^b^
Ash (%)	4.32 ± 0.27 ^c^	3.97 ± 1.16 ^d^	7.65 ± 0.33 ^a^	6.87 ± 0.20 ^b^	1.93 ± 1.25 ^e^	1.77 ± 0.09 ^f^
Carbohydrate (%)	23.21 ± 0.95 ^c^	30.43 ± 3.53 ^b^	30.12 ± 0.65 ^b^	37.75 ± 0.55 ^a^	0.0	0.0
Dietary fiber (%)	9.63 ± 0.82 ^a^	8.65 ± 0.55 ^b^	9.75 ± 0.75 ^a^	8.22 ± 0.65 ^b^	0.0	0.0
Cooking loss (%)	44.76		50.09		32.51	
Color parameters						
L*	36.11 ± 0.98 ^f^	69.51 ± 1.05 ^c^	72.57 ± 1.75 ^d^	78.25 ± 1.25 ^b^	45.45 ± 1.55 ^e^	81.05 ± 1.55 ^a^
a*	7.88 ± 0.73 ^a^	4.21 ± 0.71 ^e^	6.56 ± 1.25 ^b^	5.33 ± 1.10 ^d^	6.10 ± 1.25 ^c^	5.13 ± 1.40 ^d^
b*	6.81 ± 0.86 ^a^	4.14 ± 0.97 ^d^	6.12 ± 1.55 ^a^	4.50 ± 1.70 ^c^	6.05 ± 1.10 ^ab^	5.25 ± 1.20 ^b^
Textural properties						
Hardness (N)	2109.34 ± 768.37 ^c^	2457.85 ± 885.37 ^b^	2345.45 ± 568.75 ^b^	2687.53 ± 685.37 ^a^	1925.35 ± 235.77 ^d^	2290.55 ± 235.17 ^c^
Chewiness (N)	1477.95 ± 113.15 ^d^	1747.58 ± 233.0 ^a^	1597.99 ± 213.45 ^c^	1781.58 ± 135.0 ^a^	1323.42 ± 150.0 ^e^	1630.25 ± 203.0 ^b^
Springiness (mm)	0.90 ± 0.05 ^c^	0.94 ± 0.25 ^b^	0.85 ± 0.55 ^d^	0.90 ± 0.45 ^c^	0.94 ± 0.25 ^b^	0.99 ± 0.35 ^a^
Cohesiveness	0.52 ± 0.03 ^b^	0.77 ± 0.07 ^a^	0.59 ± 0.23 ^b^	0.70 ± 0.17 ^a^	0.40 ± 0.15 ^b^	0.55 ± 0.05 ^a^
Sensory properties						
Overall acceptability	ND	8.17 ± 1.57 ^b^	ND	8.01 ± 1.59 ^b^	ND	8.50 ± 1.59 ^a^
Texture	ND	7.90 ± 2.07 ^b^	ND	7.85 ± 1.68 ^b^	ND	8.27 ± 1.68 ^a^
Appearance	ND	8.03 ± 1.64 ^ab^	ND	7.83 ± 1.60 ^b^	ND	8.30 ± 1.60 ^a^
Aroma	ND	7.33 ± 1.73 ^b^	ND	7.27 ± 1.57 ^b^	ND	8.27 ± 1.57 ^a^
Taste	ND	7.57 ± 2.05 ^b^	ND	7.40 ± 1.70 ^b^	ND	8.10 ± 1.70 ^a^
Color	ND	7.75 ± 1.31 ^b^	ND	7.60 ± 1.52 ^b^	ND	8.20 ± 1.52 ^a^

Mean values with different superscripts in each row differ significantly (*p* < 0.05). MMMS = *Pleurotus Sajor-caju* mushroom-based minced meat substitute; CPBMM = commercial plant-based minced meat; PMM = pork minced meat.

The nutritional compositions, such as ash, protein, fat, total carbohydrate, and dietary fiber, on a dry basis (g/100 g), for both fresh and cooked MMMS, are shown in Table 10. The developed fresh MMMS had considerable amounts of protein (47.90%), fat (12.51%), ash (4.32%), carbohydrate (23.21%), and dietary fiber (9.63%). In particular, dietary protein is required for functional demands such as improving health, developing muscle, and promoting growth [77]. The consumption of PSC-mushroom-based MMMSs may substantially contribute toward the recommended dietary allowance (RDA) for protein and dietary fiber, with a recommended intake of 0.8 g of protein per kg body weight and 14 g of dietary fiber per 1000 calories of food [78]. The MMMS and CPBMM did not show any significant differences (*p* > 0.05) in terms of protein content (Table 10). However, PPM contains a significantly (*p* < 0.5) lower amount of protein. A similar trend was observed for both fresh and cooked products. Moisture and fat content was significantly higher in PMM, followed by MMMSs and CPBMM. Higher moisture and fat content makes PMM more susceptible to quick spoilage than MMMSs and CPBMM. The fat content in CPBMM was lower than in MMMSs due to formulation differences between the two products. The MMMS contained 5% canola oil whereas the CPBMM contained 1% canola oil and 1% coconut oil.

The proximate composition of PSC-mushroom-based MMSs is affected by cooking, as shown in Table 10. The raw MMMS displayed a higher (*p* < 0.05) amount of moisture content than cooked samples, which may limit its shelf-life. However, 18% moisture reduction was achieved upon cooking. This significant moisture reduction may prevent the degradation and spoilage of the cooked product, but more storage studies are required to establish the actual shelf-life. In addition, Table 10 shows that more than 14% of fat was expelled due to the cooking process. This is quite high compared to other researchers who reported a lower fat loss during cooking, such as Olagunju and Nwachukwu [79] who found that cooked beef products lost 2.74–2.90% of fat. However, only a slight reduction in protein content was observed in the present study. The reduction in protein content might be attributed to protein denaturation that occurs at high temperatures, which can also foster fat expulsion from the food matrix. Further studies should investigate how fat retention can be optimized during cooking to ensure an optimum quality. However, cooking also affects the nutritional composition of PMM and CPBMM. Table 10 shows that the cooking loss was higher in CPBMM followed by the MMMS and PMM.

The appearance and texture measurements also revealed a considerable influence of cooking on the appearance and textural properties of the MMMS. Texture analysis suggests that cooked PMM had better (*p* < 0.05) textural properties than the MMMS and CPBMM. The hardness and chewiness values were higher in cooked CPBMM than those of the MMMS and PMM. The hardness value was 2290.55 N for cooked PMM, whereas the value was 2687.53 N and 2457.85 N for cooked CPBMM and the cooked MMMS, respectively. Similar trends were observed for fresh CPBMM, the MMMS, and PMM (Table 10). However, textural parameters were considerably (*p* < 0.05) better in the MMMS than the CPBMM, with lower hardness and chewiness for both fresh and cooked samples (Table 10). The findings revealed that PMM shows better springiness than that of the MMMS, and CPBMM had the highest potential to regain its original dimension following compression. Nonetheless, the MMMS had a better springiness value than CPBMM for both fresh and cooked products. Both fresh and cooked CPBMM have less springiness, which might be due to the differences in the ingredients. However, the MMMS had lower springiness than PMM, which indicated a strong influence of the presence of starch in the formulation that contributed to changing the textural properties of the MMMS matrix [54]. A low springiness value, on the other hand, suggests that the material was plastically deformed [55]. Moreover, the maximum chewiness was observed in CPBMM. The result corresponds with the hardness value. As already discussed in the previous section, cooking resulted in an increase in lightness and a decrease in redness and yellowness values due to the breakdown of betanin from beetroot extract [66]. From this study, the color characteristics of CPBMM differed significantly (*p* < 0.05) from those of the MMMS and PMM. The lower lightness value for the MMMS was L* = 36.11 as a result of the mushroom’s inherent color, which was given a reddish tone by the beetroot extract, and the baking process may mean the development of a brown color via the Maillard reaction. The original color of the PSCs was responsible for the MMMS’s reduced brightness compared to CPBMM. The a* and b* values were higher in the MMMS than the CPBMM due to the original color of the raw materials. CPBMM is made out of soy flour (either soy flour (50 wt%) or soy protein concentrate (70 wt%)) mixed with water, sodium chloride, and additional other ingredients to form a white or faint yellow powder [80,81]. The color of the optimized MMMS was comparable to PMM, though upon heating PMM gave more of a whitish color than the MMMS. Moreover, the textural attributes of the MMMS, CPBMM, and PMM changed upon cooking, with a significant increase in hardness, chewiness, and cohesiveness (Table 10). The increase in hardness during cooking depends on a number of factors. The unfolding and aggregation of more proteins during cooking promotes more protein–protein interactions and the development of a gel-like structure. Moreover, the leaking of water and fat most likely resulted in a denser structure that was further enhanced by residual starch gelation, which both together resulted in a change in textural attributes [82].

#### 3.6.2. Sensory Properties

Although contemporary consumer trends have adopted the concepts of sustainability and wellness, the sensory properties, particularly flavor, taste, and texture of food products, are among the most important factors that customers consider when selecting whether to purchase or repurchase. The optimized MMMS from 37.5% PSC mushrooms, 12.5% chickpea flour, 0.2% beetroot extract, and 5% canola oil was designed to meet customer expectations and to mimic the qualitative features of animal-based minced meat. The sensory analysis revealed that PSC-mushroom-based MMSs containing 7.5% PSC mushrooms, 12.5% chickpea flour, 0.2% beetroot extract, and 5% canola oil have overall high acceptance, as reported by 120 untrained panelists. The result suggests that sensory properties including appearance, taste, color, texture, aroma, and overall acceptability of the developed cooked MMMS were lower (*p* < 0.05) than those of cooked PMM. The overall acceptability of the MMMS is higher than 8.0; however, texture, taste, color, and appearance were just below scores of 8.0. The best acceptability was shown by PMM rather than the MMMS and CPBMM, with scores of 8.50, 8.17, and 8.01, respectively (Table 10). From the high score mainly provided by this study, PMM shows the highest overall acceptability for attributes such as texture, appearance, aroma, taste, and color. Nonetheless, overall, the developed product (the cooked MMMS) was highly accepted and it satisfies the sensory qualities of meat products (compared with PMM). These findings support the possibility of using mushrooms to develop PB minced meat substitutes with satisfying sensory attributes and high consumer acceptance. The MMMS provided the most comparable acceptability to PMM, whereas CPBMM had the lowest acceptability rating. The aroma score was less than 8.0 for the MMMS, indicating that some volatile compounds may have negatively influenced it, such as one derived from legume ingredients that are often reported to induce off-flavors and thereby decrease aroma acceptance [83]. However, a future study has to focus on enhancing the aroma and taste of the MMMS. The outcomes from this stage can be used to inform future efforts to develop an MMMS that satisfies consumer demands. These high ratings may help to introduce such new MMMS formulations to the market as flavor and texture are key drivers in consumer decisions [84]. The result is an agreement with Sirimuangmoon et al. [85], who discovered that 50 or 80% of the meat substituted with mushrooms increased overall sensory acceptance. The use of mushrooms in the manufacturing of meat analogs, on the other hand, revealed that the organoleptic criterion for an MMMS highly depends on the overall formulation, which was also reported in the present study. For example, Nivetha et al. [86] found that a minced meat substitute with a higher sensory score can be obtained via the addition of wheat gluten, whereas the addition of paneer was less accepted. Overall, MMMS formulations containing 37.5% *w*/*w* PSC mushrooms, 12.5% *w*/*w* CF, vital wheat gluten (4.8%, *w*/*w*), distilled water (28%, *w*/*w*), soy protein isolate (10%, *w/w),* canola oil (5%, *w*/*w*), beetroot extract (0.2%, *w*/*w*), and yeast extract (2%, *w/w)* show promising texture and flavor profiles, which may lead to a higher consumer adoption of meat alternatives.

#### 3.6.3. Protein Patterns of MMMS, CPBMM, and PMM

SDS-PAGE was used to determine the protein pattern of a PSC-mushroom-based MMS, and we compared it with PMM and CPBMM. The SDS-PAGE profiles of a PSC-mushroom-based MMS (a), CPBMM (a), and PMM (a), are shown in Figure 4. For every sample, three major bands were observed at ~65, 100, and ~130 kDa, with likely corresponding patterns for the protein composition of the three samples.

Blanching and cabinet drying of mushrooms may denature protein and change their molecular weight profile distribution [18]. Albumins, globulins, glutelin-like materials, glutelins, prolamins, and prolamin-like materials were the major protein fractions in mushrooms. The majority of soy protein is made up of two common proteins, 7S β-conglycinin (about 40% of total protein) and 11S glycinin (about 30% of total protein) [87]. Gliadin and glutenin, which especially have typical properties that set them apart as being unique from other plant proteins, make up around 85% of the proteins in wheat gluten [87]. When mixed with water, both of them help to form a viscoelastic matrix typical of bread dough and also help to develop the disulfide bonds that give textured plant proteins a fibrous structure [88]. Along with soy protein and wheat gluten, chickpeas are another protein source in MMMSs. A total of 32% of the protein in chickpeas is legumin, which has a protein pattern with molecular weights of 75 and 70 kDa. Vicilin was a higher soy protein than legumin, which is larger in size and contains more sulfide groups. Despite its lower content, it is an important component of protein texturization because of the sulfide groups that it contributes [69].

## 4. Conclusions

This study described a novel method for producing a mushroom-based minced meat substitute using *Pleurotus sajor-caju* as a main ingredient along with chickpea flour, isolated soy protein, and vital wheat gluten as protein sources. The base formulation suggests that a MMMS with PSC mushrooms shows considerable amounts of protein and better sensory acceptance. Chickpea flour was used to improve the textural properties by mixing it with PSC mushrooms in ratios of 0:50, 12.5:37.5, 25:25, 37.5:12.5, and 50:0 (*w/w).* Textural and sensory attributes suggest that PSC mushrooms to chickpeas in a ratio of 37.5:12.5 shows higher acceptability of the MMMS with the protein content up to 47%. Canola oil (5%, *w/w)* shows better consumer acceptability with maximum overall acceptability, appearance, juiciness and texture scores. Beetroot extract (0.2% *w/w)* had higher consumer acceptance, showing higher scores for overall acceptability, appearance, and fresh and cooked aroma and color. Overall, an optimum formulation containing 37.5% *w*/*w* PSC mushrooms, 12.5% *w*/*w* chickpea flour, 5% *w*/*w* canola oil, and 0.2% *w*/*w* beetroot extract was selected for the production of the MMMS based on nutritional, textural, and sensory characteristics. These results show that it is possible to formulate a nutritious meat analog with high consumer acceptance based on mushrooms. The development of the MMMS is anticipated to broaden the uses of mushrooms, expand the meat alternative portfolio, and respond to consumers’ expectations, as well as the sustainability of the food supply in the future. Storage stability, bioavailability, and in vivo analysis of allergenicity of the MMMS should be the focuses of further research.

This study described a novel method for producing a mushroom-based minced meat substitute using *Pleurotus sajor-caju* as a main ingredient along with chickpea flour, isolated soy protein, and vital wheat gluten as protein sources. The base formulation suggests that a MMMS with PSC mushrooms shows considerable amounts of protein and better sensory acceptance. Chickpea flour was used to improve the textural properties by mixing it with PSC mushrooms in ratios of 0:50, 12.5:37.5, 25:25, 37.5:12.5, and 50:0 (*w/w).* Textural and sensory attributes suggest that PSC mushrooms to chickpeas in a ratio of 37.5:12.5 shows higher acceptability of the MMMS with the protein content up to 47%. Canola oil (5%, *w/w)* shows better consumer acceptability with maximum overall acceptability, appearance, juiciness and texture scores. Beetroot extract (0.2% *w/w)* had higher consumer acceptance, showing higher scores for overall acceptability, appearance, and fresh and cooked aroma and color. Overall, an optimum formulation containing 37.5% *w*/*w* PSC mushrooms, 12.5% *w*/*w* chickpea flour, 5% *w*/*w* canola oil, and 0.2% *w*/*w* beetroot extract was selected for the production of the MMMS based on nutritional, textural, and sensory characteristics. These results show that it is possible to formulate a nutritious meat analog with high consumer acceptance based on mushrooms. The development of the MMMS is anticipated to broaden the uses of mushrooms, expand the meat alternative portfolio, and respond to consumers’ expectations, as well as the sustainability of the food supply in the future. Storage stability, bioavailability, and in vivo analysis of allergenicity of the MMMS should be the focuses of further research.

## Figures and Tables

**Figure 1 foods-12-02094-f001:**
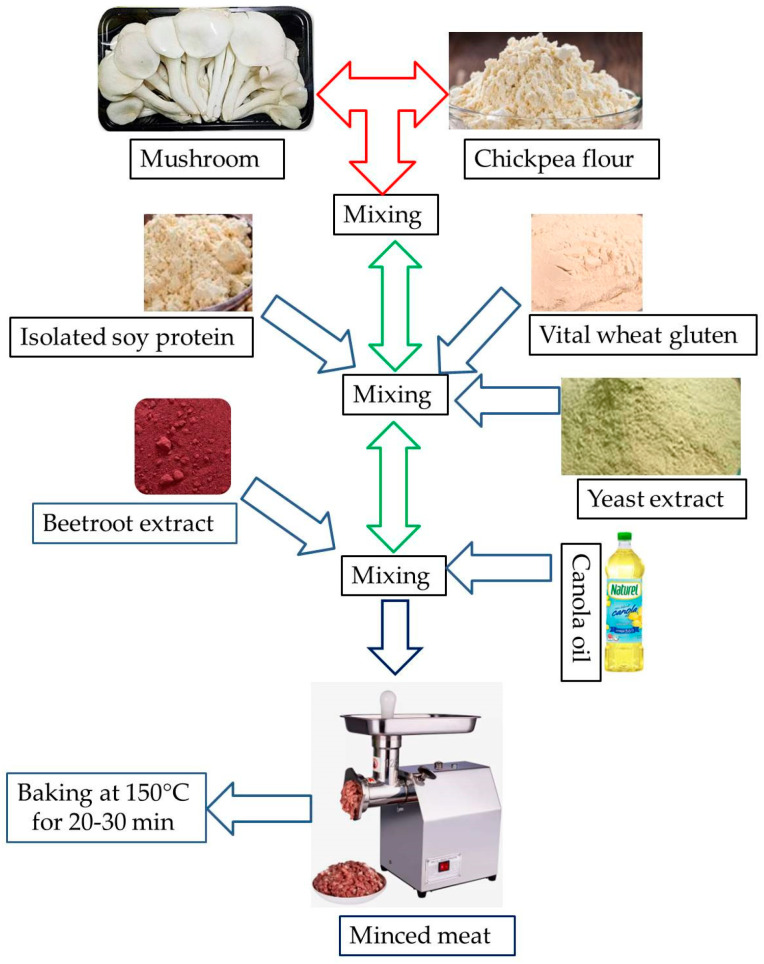
Processing of *Pleurotus sajor-caju* mushroom-based minced meat substitute.

**Figure 2 foods-12-02094-f002:**
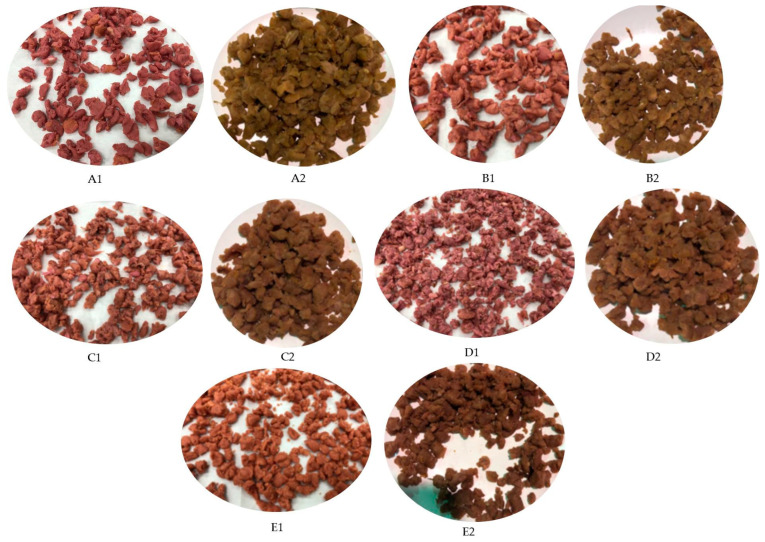
Effect of different concentrations of beetroot extracts on the appearance of fresh and cooked *Pleurotus*
*sajor*-*caju*-based minced meat substitute. (**A1**) = fresh MMMS (0.2% *w*/*w* beetroot extract); (**A2**) = cooked MMMS (0.2% *w*/*w* beetroot extract); (**B1**) = fresh MMMS (0.4% *w*/*w* beetroot extract); (**B2**) = cooked MMMS (0.4% *w*/*w* beetroot extract); (**C1**) = fresh MMMS (0.6% *w*/*w* beetroot extract); (**C2**) = cooked MMMS (0.6% *w*/*w* beetroot extract); (**D1**) = fresh MMMS (0.8% *w*/*w* beetroot extract); (**D2**) = cooked MMMS (0.8% *w*/*w* beetroot extract); (**E1**) = fresh MMMS (1.0% *w*/*w* beetroot extract); (**E2**) = cooked MMMS (1.0% *w*/*w* beetroot extract).

**Figure 3 foods-12-02094-f003:**
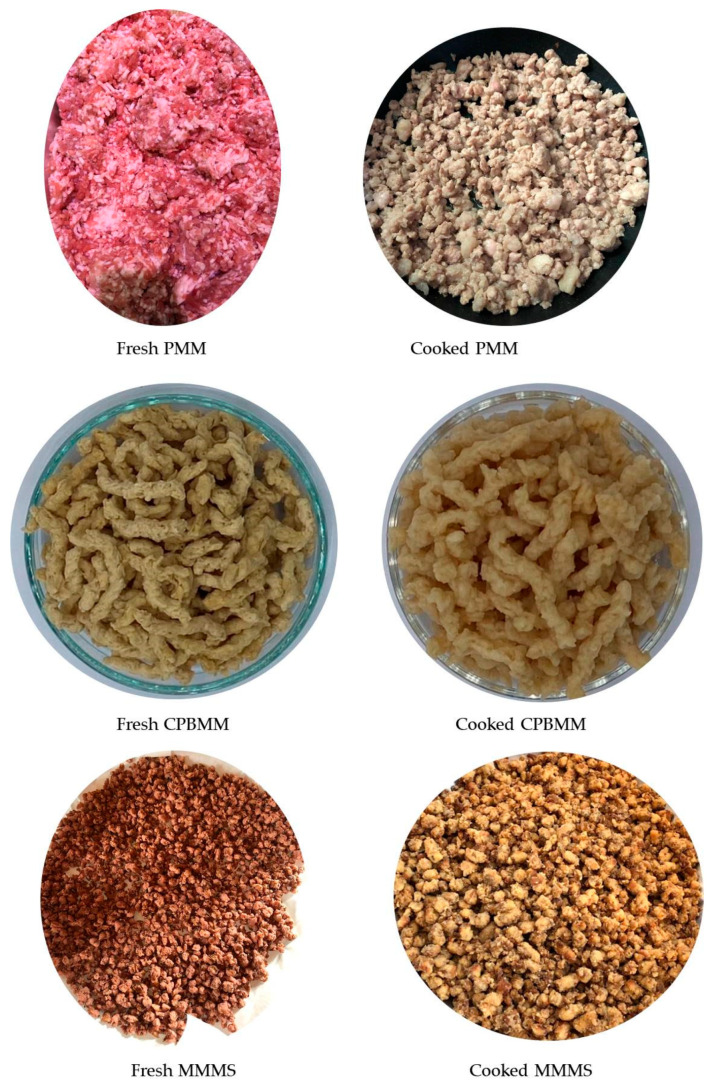
Appearance of the PMM, CPBMM, and MMMS. PMM = pork minced meat; CPBMM = commercial plant-based minced meat; MMMS = *Pleurotus Sajor-caju* minced meat substitute.

**Figure 4 foods-12-02094-f004:**
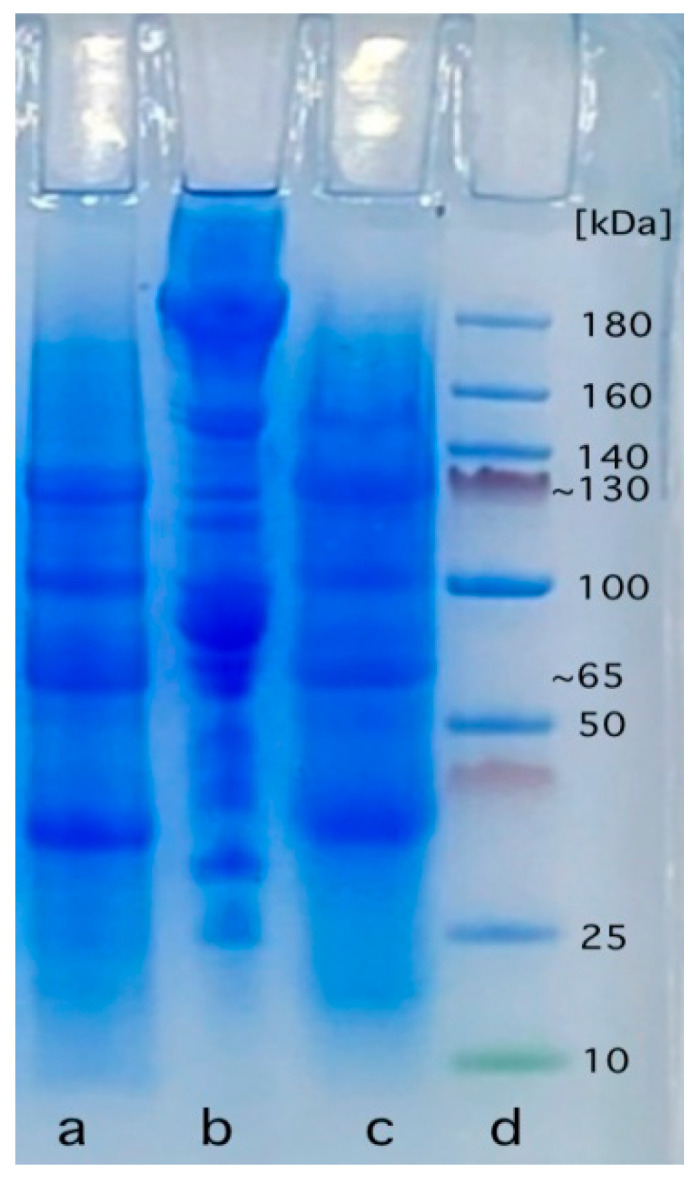
Electrophoresis patterns of protein profiles of different minced meat substitutes: (**a**) *Pleurotus sajor-caju* mushroom-based minced meat substitute; (**b**) pork minced meat; (**c**) commercial plant-based minced meat; (**d**) standard marker.

**Table 1 foods-12-02094-t001:** Base formulation of mushroom-based minced meat substitutes.

Ingredients	%, by Weight
*Pleurotus sajor-caju*	25.0
Chickpea flour	25.0
Distilled water	28.0
Isolated soy protein	10.0
Vital wheat gluten	4.8
Beetroot extract	0.2
Canola oil	5.0
Yeast extract	2.0

**Table 2 foods-12-02094-t002:** Effect of different concentrations of chickpea flour on mushroom-based minced meat substitutes.

Ingredients (%, by Weight)	Minced Meat Substitutes				
	A	B	C	D	E
*Pleurotus sajor-caju*	0	12.5	25	37.5	50
Chickpea flour	50	37.5	25	12.5	0
Distilled water	28				
Isolated soy protein	10				
Vital wheat gluten	4.8				
Beetroot extract	0.2				
Canola oil	5				
Yeast extract	2				
Total	100				

A = *Pleurotus sajor-caju:* chickpea flour (0:50); B = *Pleurotus sajor-caju:* chickpea flour (12.5:37.5); C = *Pleurotus sajor-caju:* chickpea flour (25:25); D = *Pleurotus sajor-caju:* chickpea flour (37.5:12.5); E *Pleurotus sajor-caju:* chickpea flour (50:0).

**Table 3 foods-12-02094-t003:** Effect of different concentrations of beetroot extract on mushroom-based minced meat substitutes.

Ingredients (%, by Weight)	Minced Meat Substitutes					
	C	F	G	H	I	J
*Pleurotus sajor-caju*	37.5					
Chickpea flour	12.5					
Distilled water	28					
Isolated soy protein	10					
Vital wheat gluten	5.0	4.8	4.6	4.4	4.2	4.0
Beetroot extract	0	0.2	0.4	0.6	0.8	1.0
Canola oil	5					
Yeast extract	2					
Total	100					

C = Control (without beetroot extract); F = 0.2% (*w*/*w*) beetroot extract; G = 0.4% (*w*/*w*) beetroot extract; H = 0.6% (*w*/*w*) beetroot extract; I = 08% (*w/w)* beetroot extract; J = 1.0% (*w/w)* beetroot extract.

**Table 4 foods-12-02094-t004:** Effect of different concentrations of canola oil on mushroom-based minced meat substitutes.

Ingredients (%, by Weight)	Minced Meat Substitutes					
	C	K	L	M	N	O
*Pleurotus sajor-caju*	37.5					
Chickpea flour	12.5					
Distilled water	33	32	31	30	29	28
Isolated soy protein	10					
Vital wheat gluten	4.8					
Beetroot extract	0.2					
Canola oil	0	1	2	3	4	5
Yeast extract	2					
Total	100					

C = Control (without canola oil); K = 1% (*w*/*w*) canola oil; L = 2% (*w*/*w*) canola oil; M = 3% (*w*/*w*) canola oil; N = 4% (*w*/*w*) canola oil; O = 5% (*w*/*w*) canola oil.

**Table 6 foods-12-02094-t006:** Moisture, protein content, and sensory attributes of *Pleurotus sajor-caju* mushroom minced meat substitute base formulation.

Properties		PSC MMMS
Moisture (%)		28.39 ± 0.17
Protein (% db)		41.99 ± 0.55
Sensory attributes	Overall acceptability	6.43 ± 1.80
	Appearance	6.80 ± 1.47
	Color	6.78 ± 1.74
	Aroma	5.93 ± 1.68
	Taste	5.91 ± 1.81
	Texture	6.43 ± 1.82

PSC MMMS = *Pleurotus sajor-caju* mushroom-based minced meat substitute.

**Table 7 foods-12-02094-t007:** Effect of different concentrations of chickpea flour on the properties of *Pleurotus sajor-caju* mushroom-based minced meat substitute.

Properties		PSC Mushroom: Chickpea Flour (by Weight)					Pork Minced Meat
		0:50	12.5:37.5	25:25	37.5:12.5	50:0	
Moisture (%)		12.30 ± 0.06 ^e^	12.99 ± 0.12 ^d^	13.74 ± 0.29 ^c^	14.86 ± 0.55 ^b^	16.16 ± 0.06 ^ab^	60.10 ± 0.25 ^a^
Protein (% d.b.)		34.29 ± 0.17 ^d^	37.74 ± 0.23 ^c^	39.69 ± 0.43 ^b^	47.03 ± 0.28 ^a^	47.59 ± 0.96 ^a^	20.17 ± 0.70 ^c^
Textural properties	Hardness (N)	9441.01 ± 1683.09 ^a^	3668.28 ± 373.81 ^b^	2721.81 ± 838.41 ^b^	2610.23 ± 292.59 ^b^	1983.35 ± 711.42 ^c^	1925.35 ± 235.77 ^c^
	Chewiness (N)	3422.55 ± 103.09 ^a^	1347.78 ± 273.41 ^b^	1220.32 ± 138.41 ^b^	1171.32 ± 192.90 ^b^	789.84 ± 173.41 ^c^	1323.42 ± 150 ^b^
	Springiness (mm	0.65 ± 0.04 ^e^	0.76 ± 0.19 ^d^	0.86 ± 0.06 ^bc^	0.88 ± 0.07 ^ab^	0.90 ± 0.11 ^a^	0.94 ± 0.25 ^a^
	Gumminess	826.99 ± 91.31 ^a^	791.45 ± 90.29 ^a^	660.54 ± 456.18 ^b^	673.47 ± 88.52 ^ab^	775.54 ± 80.97 ^a^	615.66 ± 75.20 ^c^
	Cohesiveness	0.35 ± 0.04 ^c^	0.50 ± 0.10 ^abc^	0.57 ± 0.05 ^ab^	0.45 ± 0.15 ^bc^	0.63 ± 0.06 ^a^	0.40 ± 0.15 ^bc^
Sensory attributes	Overall acceptability	4.44 ± 1.64 ^e^	5.09 ± 1.65 ^d^	5.47 ± 1.86 ^d^	7.24 ± 0.89 ^b^	6.24 ± 1.60 ^c^	8.05 ± 1.59 ^a^
	Appearance	5.24 ± 1.74 ^c^	5.18 ± 1.64 ^c^	5.53 ± 1.86 ^c^	7.21 ± 1.17 ^b^	6.00 ± 1.78 ^c^	8.10 ± 1.68 ^a^
	Aroma	5.53 ± 1.50 ^e^	6.06 ± 1.40 ^d^	6.77 ± 1.27 ^c^	7.11 ± 1.25 ^b^	7.04 ± 1.32 ^b^	8.30 ± 1.60 ^a^
	Texture	3.03 ± 1.70 ^f^	4.26 ± 1.94 ^e^	5.41 ± 1.97 ^d^	7.65 ± 1.00 ^b^	6.24 ± 1.92 ^c^	8.07 ± 1.57 ^a^

Mean values with different superscripts in each row differ significantly (*p* < 0.05).

**Table 8 foods-12-02094-t008:** Effects of canola oil concentrations on sensory properties of *Pleurotus sajor-caju* mushroom-based minced meat substitutes.

Canola Oil (%, *w/w)*	Overall Acceptability	Appearance	Juiciness	Aroma	Texture
0	4.77 ± 1.01 ^e^	5.07 ± 1.20 ^d^	4.15 ± 1.55 ^d^	4.29 ± 0.97 ^e^	4.10 ± 1.33 ^e^
1	6.37 ± 1.21 ^d^	6.60 ± 1.52 ^b^	5.46 ± 1.65 ^c^	5.60 ± 1.53 ^d^	6.26 ± 1.40 ^c^
2	6.46 ± 1.36 ^c^	6.60 ± 1.50 ^b^	5.57 ± 1.54 ^bc^	6.05 ± 1.50 ^c^	6.26 ± 1.68 ^c^
3	6.46 ± 1.30 ^c^	6.60 ± 1.60 ^b^	5.54 ± 1.68 ^bc^	6.70 ± 1.50 ^bc^	6.14 ± 1.51 ^d^
4	6.80 ± 1.54 ^b^	6.51 ± 1.71 ^c^	5.06 ± 1.87 ^ab^	6.81 ± 1.20 ^b^	6.69 ± 1.77 ^b^
5	6.97 ± 1.27 ^a^	6.89 ± 1.63 ^a^	6.60 ± 1.72 ^a^	7.21 ± 1.00 ^a^	6.74 ± 1.62 ^a^

Mean values with different superscripts in each row differ significantly (*p* < 0.05).

**Table 9 foods-12-02094-t009:** Color attributes and sensorial properties of *Pleurotus sajor-caju* mushroom-based minced meat substitutes using different levels of beetroot extract.

Properties			Concentration of Beetroot Extract (%, *w*/*w*)				
		0	0.2	0.4	0.6	0.8	1.0
Fresh MMMS color							
Whiteness		0.0	30.55 ± 0.55	27.85 ± 1.55	25.45 ± 0.90	25.29 ± 1.11	23.79 ± 2.01
ΔE		0.0	20.76 ± 1.03	18.03 ± 1.25	17.20 ± 1.35	16.10 ± 0.95	15.70 ± 0.75
L*		65.75 ± 0.95 ^d^	38.89 ± 0.33 ^c^	34.79 ± 0.03 ^b^	33.55 ± 0.56 ^b^	33.40 ± 0.49 ^b^	31.34 ± 0.93 ^a^
a*		5.13 ± 0.55 ^a^	7.83 ± 0.09 ^b^	7.74 ± 0.00 ^b^	8.01 ± 0.81 ^bc^	8.27 ± 0.32 ^c^	8.39 ± 0.07 ^c^
b*		9.20 ± 0.25 ^e^	6.87 ± 0.06 ^d^	4.40 ± 0.01 ^c^	3.45 ± 0.15 ^bc^	2.94 ± 0.45 ^b^	2.41 ± 0.23 ^a^
Cooked MMMS color							
Whiteness		0.0	65.25 ± 1.03	62.47 ± 0.98	61.72 ± 0.85	61.25 ± 2.03	60.12 ± 2.20
ΔE		0.0	45.90 ± 1.15	44.40 ± 1.95	43.85 ± 1.22	43.35 ± 1.75	43.25 ± 1.55
L*		87.70 ± 0.90 ^d^	85.32 ± 0.71 ^c^	82.12 ± 0.55 ^b^	81.27 ± 0.47 ^a^	80.76 ± 0.58 ^a^	80.38 ± 0.21 ^a^
a*		3.45 ± 0.25 ^a^	4.37 ± 0.66 ^ab^	4.45 ± 0.11 ^b^	4.93 ± 0.82 ^c^	4.43 ± 0.94 ^b^	4.76 ± 0.88 ^b^
b*		5.79 ± 0.35 ^d^	4.65 ± 0.57 ^c^	4.43 ± 0.28 ^c^	3.13 ± 0.90 ^b^	2.38 ± 1.17 ^a^	2.13 ± 0.35 ^a^
Sensory attributes	Overall acceptability	4.25 ± 1.02 ^cd^	6.85 ± 1.33 ^a^	4.62 ± 1.46 ^c^	6.56 ± 1.60 ^a^	5.50 ± 1.66 ^b^	5.50 ± 1.78 ^b^
	Appearance	4.75 ± 1.70 ^c^	7.15 ± 1.40 ^a^	5.21 ± 1.63 ^b^	6.62 ± 1.52 ^a^	5.62 ± 1.65 ^b^	5.59 ± 1.84 ^b^
	Fresh texture	7.05 ± 0.92 ^c^	7.93 ± 1.20 ^a^	7.85 ± 1.50 ^a^	7.78 ± 1.70 ^a^	7.61 ± 130 ^ab^	7.55 ± 0.09 ^ab^
	Cooked texture	6.95 ± 1.30 ^c^	7.75 ± 1.50 ^a^	7.70 ± 1.75 ^a^	7.55 ± 1.90 ^a^	7.41 ± 1.25 ^ab^	7.35 ± 1.10 ^ab^
	Fresh aroma	6.15 ± 0.75 ^d^	7.22 ± 1.75 ^a^	6.89 ± 1.50 ^b^	6.72 ± 1.85 ^bc^	6.59 ± 1.90 ^bc^	6.41 ± 1.33 ^cd^
	Cooked aroma	6.35 ± 1.55 ^c^	7.64 ± 1.60 ^a^	7.05 ± 1.50 ^b^	6.82 ± 1.70 ^bc^	6.64 ± 1.55 ^bc^	6.58 ± 1.64 ^bc^
	Fresh color	3.95 ± 1.05 ^d^	6.82 ± 1.55 ^a^	3.79 ± 1.88 ^d^	5.62 ± 1.78 ^b^	5.09 ± 1.94 ^bc^	4.41 ± 1.93 ^cd^
	Cooked color	4.02 ± 1.72 ^c^	6.94 ± 1.46 ^a^	4.59 ± 1.20 ^c^	7.12 ± 1.60 ^a^	5.74 ± 1.62 ^b^	5.18 ± 1.64 ^bc^

Mean values with different superscripts in each row differ significantly (*p* < 0.05).

## Data Availability

The datasets generated for this study are available on request to the corresponding author.

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
