# Peer review of "Mushroom–Legume-Based Minced Meat: Physico-Chemical and Sensory Properties"

_foods, 2023, doi:10.3390/foods12112094_

Round 1

Reviewer 1 Report

I reviewed the manuscript titled “Mushroom-Legume-Based Minced Meat: Physico-chemical and Sensorial Properties. Although the study focused on development of minced meat alternatives, authors failed to compare the developed product with real animal-based minced meat. Without comparing the developed product with real animal-based minced meat, there is no possibility to claim that the developed product is an alternative to animal-based meat. Based on the evaluated parameters, it is not an alternative to animal-based meat as it has many differences and no comparison. Based on these observations, this manuscript is not suitable for publication consideration in this journal.

Line 23: Pleurotus Sajorcaju (SC). How the abbreviation is SC. It is supposed to be PSC, which might be in line with scientific names and easy to follow.

Line 23  “s” must be lower-case in Sajorcaju.

Abstract must be revised based on the study findings. Authors explained a lot on methodology

Results must be presented and conclusions must be derived appropriately

Objectives in the abstraction section were not marching with the objectives in the Introduction section. I suggest authors revise the objectives in the abstract section.

Line 47: ref number 15 appeared right after ref 5. This is not the correct format. All ref must come in sequence; for example, 1,2,3,4,5,6,7, etc.,

METHODOLOGY

This section is appropriate; however, Line 209: at least 46, 46, 34, 35, and 120 - untrained panelists…. What do the numbers mean?

Table 1: it is not suggested to be mentioned as Sajor-caju. I suggest to keep P. sajor-caju

All scientific names must be in Italics

Authors developed a minced meat alternative. However, authors failed to compare this product with animal minced meat to know how similar it is with animal meat. Authors must perform the set of experiments to compare the animal minced meat and plant-based alternative (chickpea addition and extract addition). Without performing the test, we cannot claim that the meat alternatives can be used to replace animal-based meat.

Table 3: there is no positive and negative control. I suggest authors perform few experiments. For example, control, animal minced meat (with chick-pea coating and beetroot coating; without both coatings; with oil addition and without oil addition); marketed plant-based minced meat (if available in Thailand). Based on my experience, many plant-based meat products are available in Thailand. Without performing these experiments, It may not contribute to the novelty in this field and thus in this journal for publication.

In many places, authors compared the plant-based meat. However, it is better to compare with real animal-based minced meat to understand how similar it is in texture and other quality aspects.

Why authors did not keep control or positive control (i.e., animal-based meat).

There is no control for textural studies.

Overall, authors must perform sensory analysis by keeping animal-based meat as positive control and compare with developed products.

Conclusions must be revised

References must be cross checked

Author Response

We try to revise this article according to the reviewer comments and suggestions. The correction was made by track changes, red color & yellow color highlighted text for reviewer 1 

Reviewer 2 Report

After reading the manuscript  "Mushroom-Legume-Based Minced Meat: Physico-chemical and Sensorial Properties"  I realized that the manuscript showed in some parts the scientific rigour wanted, but in other parts I have missed it.

The authors have presented critical evaluation only in some paragraphs.

The references are not exactly current, besides the objective should be  improved.

Thats why I have written some suggestions below in an attempt to improve the paper. 

L.37- I missed the reference [1] in the Introduction

L.43- You use " nowadays" but the reference is from 2019. This subject has advanced a lot in the last 2 years, I think it is interesting that it is reviewed.

L.47- Reference [15] here ? , please check Journals guide for authors.

L.49- Take care with "recent" another tricky word, just like "nowadays"

L.61- "Recent years have seen an increase in interest in..."  Check English, please.

L.65- Paragraph ?

L.68- I think you need to reorganise the sentences/ideas about fungi /mushrooms... so the text would flow better and avoiding back and forth in the introduction of the paper.

L.71- " Nowadays" again- same problem [26,27,28,29] => including a reference from 2004, no way nowadays, please think about it.

L.83- Sensory instead of organoleptic.

L.84-89 - It has become very repetitive. Review the objective.  Remember the title mentions  "Physico-chemical and Sensorial Properties"  by the way, prefer sensory instead of sensorial.

L.103- Was SC defined previously in Introduction or Material and Methods? remember : some people read only abstract, others go staight to the Introduction, so please. insert SC in line 92

L.123- a table with this information would be perfect. L.113. I got confused here 25% chickpea or (0, 12.5, 25, 37.5, 50% w/w) of chickpea. Definitely a table would be more informative.

L.128- I could not appreciate the supplementary material, the last page was page 22. Anyway, I think the main table about the formulations should not be a supplementary document. It is very important for other researchers to know it for future projects.

L.135- " 2019 AOAC " - Correct, please.

L.186- I only found about the repetitions below the tables, it seems to me that it should be in the Material and methods. The texture was done in triplicate also? Could that have been a bias? 

L.209- "46, 46, 34, 35, and 120" were those sessions ? I did not understand, i am sorry. Could you be clearer.  What was the profile of the assessors ?  How many assessors ? How many men/women ? age of the assessors ? Are the assessors usually consumers of this product  ? 

L.210- "optimum concentration of beetroot extract" - This is not in your objectives. Please, take care.

L.212 and 213- your attributes are different from table 2.

L.247- "Morphological characteristics and nutritional properties of Sajor-caju mushroom" - Is it in your objectives, title  ? Mushroom or beetroot extract or minced meat, please focus on minced meat.

L.340- What about the other chemical analysis? I prefer this way. The beetroot extract could be another paper.

L.385- "Effect of canola oil on sensory characteristics" was this an objective ? Focus on the objective. 

L. 415-   "Effect of canola oil on sensory characteristics" was this an objective ? Focus on the objective. 

L.420-  Effect of beetroot extract on color and sensory characteristics - was this an objective ? Focus on the objective. 

 L.453- Please check your sensory attributes in Material and methods.

L.511-Figure 2- It seems to me that the authors ran away from the objectives with this figure and focused on the "beetroot extract". You need to think about what the objective of the work is, look at the title and the conclusion of the study.

L.531- It got confusing which analysess were with hamburger and which were with minced meat cooked. Make it clearer. Decide your real objective.

Moderate editing of English language

Author Response

All the reviewer’s comments and suggestion are listed and responded accordingly in separate files as attached. The correction was made by track changes & red color for reviewer 2.  

Round 2

Reviewer 1 Report

Authors are now performed experiments and included the data. The quality of the manuscript is now improved. In my opinion, this version can be accepted for possible publication consideration.

Author Response

Thank you very much for your suggestion and effort.  

We appreciated 

Reviewer 2 Report

Authors,

After another evaluation of the manuscript, I realized a great improvement in the quality of the paper. The authors have accepted almost all of my requests. This paper still has room for improvement. Some questions to think about them:

Why extract evaluation is not in your title ?  Since your objective is "... CF, beetroot extract and canola oil, (ii) optimize the formulation using 151 PSC mushroom, CF, beetroot extract and canola oil, ..."

authors,  I honestly think that the extracts part didn't need to be in this paper.

The number of assessors for the sensory analysis was deleted. I didn't understand the 50 assessors, in the previous version it said "46, 46, 34, 35, and 120". Please, correct and insert the total number correctly.age of the assessors? 18-75 ? wow ... you have teenagers, adults and elderly... Could be a bias ?

Are the assessors usually consumers of this product? 

This version came to me very misconfigured, so I couldn't evaluate the tables well. I ask you to review and check everything in a word version. 

A final revision by a native English-speaking would be very important.

Minor editing of English language required

Author Response

Thank you very much for your invaluable suggestion and comment. Please see the file attached for correction point-by-point
